# The Effects and Mechanisms of the Rural Homestead System on the Imbalance of Rural Human–Land Relationships: Evidence from the Yangtze River Delta Urban Agglomeration in China

**Yuan Yi** [1] , **Kaifeng Duan** [2] , **Fang He** [1] **and Yuxuan Si** [1,*]

1   School of Economics and Management, Tongji University, Shanghai 200092, China;
    1810045@tongji.edu.cn (Y.Y.); heyoufang@tongji.edu.cn (F.H.)
2   School of Economics and Management, Fuzhou University, Fuzhou 350108, China; kefee920729@fzu.edu.cn
*   Correspondence: 2310123@tongji.edu.cn; Tel.: +86-18017496839

**Abstract:** The imbalance of rural human–land relationships has become a notable problem in China's urbanization process. The dual urban–rural system is widely regarded as the crucial factor contributing to this problem in China. Although the significance of institutional forces has been substantially recognized, the rural homestead system seems to be generally under-evaluated in this issue. Most of the previous literature focuses on the dual household registration system, while the effects and the detailed mechanisms of the rural homestead system on human–land relationships lack depth in research. The objective of this research is to help fill this gap in the literature on the complex effects and the detailed mechanisms of the rural homestead system on rural human–land relationships. In view of this, this paper establishes a conceptual framework on the basis of land function theory and public domain of property rights theory and proposes two mechanism hypotheses: one is the land attachment mechanism of farmers' rights and interests (LAM), the other is the land finance preference mechanism of local governments (LFPM). Then, this article examines them empirically using the panel model with the data of 41 cities from 2010 to 2021 in the Yangtze River Delta of China. The main conclusions are as follows: (1) LAM promotes the imbalance of rural human–land relationships due to the attachment of farmer's social security rights and property expectant interests to the rural homesteads; (2) LFPM drives the imbalance of rural human–land relationships, owing to both the preference of land transfer revenue and the exclusion of rural migrants' citizenship financial cost on local governments; (3) the moderating effects suggest that LFPM can strengthen the effect of LAM, and the spatial Durbin model results show that both LAM and LFPM have spatial spillover effects. It is hoped that the findings will provide a reference for deepening the rural homestead system reform.

**Keywords:** rural homestead system; rural human–land relationship; land attachment mechanism; land finance preference mechanism; panel model; spatial Durbin model

## 1. Introduction

Since the reform and opening up in the 1980s, China's urbanization has entered a phase of rapid development, and the urban–rural mobility of production factors such as population, land, and capital has laid the foundation for China's rapid economic growth [1]. After more than 40 years of urbanization, the landscape of China's rural areas has changed enormously, along with the transformation of rural socio-economic patterns [2]. At the same time, as the permanent resident population in rural areas continues to decline, the area of rural settlements is increasing rather than decreasing, and the phenomenon of disorderly rural settlements expansion, hollowing villages, and idle homesteads is very serious [3,4]. From 2010 to 2020, the national rural resident population went down from 662 million to 510 million (a decrease of 23%) [5], while the area of rural settlements in the same period went from 185,900 km$^2$ to 218,600 km$^2$ (an increase of 18%), and the per capita area of rural settlements expanded from 281 m$^2$ to 429 m$^2$ (an increase of 53%) [6].

This reflects the prominent problem of the imbalance in rural human–land relationships at present [7,8].

China's urban–rural dual land system is regarded as the main institutional cause of the above problem [9]. The current rural homestead system, established during the planned economy, is identity based and welfare oriented and has played a positive role in maintaining long-term stability in rural society. However, along with the large number of rural residents moving to cities and towns, the shortcomings of the existing homestead system were gradually exposed. The system has severely restricted the transfer of the right to use homesteads and caused unfair distribution of land value-added benefits between the government and farmers [10,11], hindering the reasonable flow of rural land. Because they lack the legal means to transfer land, numerous rural migrants, who work and live in cities most of the time, still keep their hometown unused rural residences on hand. As time goes on, the problem of inefficient use of rural land and uncoordinated rural human–land relationships become increasingly serious.

As the existing system was not adapted to the needs of high-quality urbanization, calls for land system reform (LSR) are growing louder [12]. Clarity of property rights, tenure security, efficient use of resources, and coordination of human–land relationships are the core demands for LSR [13]. Since 2015, the Chinese government has carried out a pilot reform in 33 counties (municipalities and districts) focusing on the reform of rural land expropriation, the market entry of rural collective construction land, and the rural homestead system. The amended Land Administration Law of China in 2019 stipulates that the State allows rural villagers who have settled in cities to voluntarily withdraw from their homesteads with compensation and encourages rural collective economic organizations and their members to revitalize idle homesteads and houses. This reflects the policy judgment of the central government that the current rural land system hinders the coordinated development of rural human and rural land, as well as showing its determination to promote LSR to solve the problems of unbalanced and insufficient development in rural areas [14].

There has been a growing number of studies on China's dual urban–rural system in recent years. Although the system barrier to the imbalanced rural human–land relationship has been widely acknowledged, few studies have examined the detailed mechanisms of the rural homestead system from the perspective of agent decision behavior, namely how the rural homestead system affects farmers' land use behavior and governments' land finance behavior, and how the agent decision behaviors bring about the current rural human–land relationship. Moreover, the bonds between the rural homestead system, the household registration system, and the social security system and the effect of these bonds on rural human–land relationships have not been well explored. With regard to the research methods, most of the above studies were limited to theoretical analysis. A few quantitative studies were limited to surveys about the willingness of migrants, neglecting the difference between willingness and final decision-making, as well as lacking macro-level statistical analysis. The objective of this research is to help fill this gap in the literature on the complex effects and the detailed mechanisms of the rural homestead system on rural human–land relationships, and it is hoped that the findings will provide a reference for deepening the rural homestead system reform, improving the efficiency of rural land use, and promoting high-quality urbanization.

In view of this, this paper focuses on the following two questions: (1) Do the land attachment mechanism of farmers' rights and interests (LAM) and the land finance preference mechanism of governments (LFPM) encourage the imbalance of rural human–land relationships? (2) What is the detailed influence path of LAM and LFPM? Thereby, on the basis of land property rights theory and land function theory, this paper analyses LAM and LFPM in theory and further conducts an empirical study to testify LAM and LFPM through panel regression with the data of 41 cities in the Yangtze River Delta from the 2010–2021 period. We draw the conclusion that the current rural homestead system is the major trigger for bringing about the imbalanced rural human–land relationship through

LAM and LFPM and highlight the significance of deepening the linkage reforms of the dual urban–rural land system, the household registration system, and the social security system.

The rest of this paper is organized as follows. Section 2 reviews the literature. Section 3 shows the research hypotheses. Section 4 presents the materials and methods. The results are presented in Section 5. Finally, Section 6 shows the discussions and draws the conclusions.

## 2. Literature Review

The related research mainly focused on two aspects. One is the state and drivers of rural human–land relationships, and the other one is the features and flaws of the current rural homestead system.

### 2.1. Research on the State and Drivers of Rural Human–Land Relationships

Previous research assessed the state of rural human–land relationships by measuring indicators such as the per capita area of residential land [15], land-use efficiency [16], the coupled coordination degree between rural residents and rural residential land [17], and Tapio's decoupling index [18]. Most studies found that the per capita area of homesteads exceeded the standard in most areas and suggested that there is a serious problem of imbalance in human–land relationships (namely, fewer people, more land) [19]. Some research further explored the drivers of the current uncoordinated state, including economic development [20], the urbanization of rural settlements [21], the land finance incentives of local governments [22,23], system barriers such as ambiguous property rights [24], restrictions to the transfer of rural homesteads, and the inadequate rural social security system [25]. Among them, the urban–rural dual land system is seen as the key cause [18], and the land finance incentives of local governments are regarded as an important trigger [26,27].

### 2.2. Research on the Features and Flaws of Rural Homestead System

In accordance with the Constitution, the Land Management Law, and the Property Law of China, scholars have investigated the features and flaws of China's rural homestead system [28,29]. Their main conclusions are the following: (1) Acquisition system: Free acquisition, long-term use, and ownership belong to the collective [30]. This intensifies the problem of overoccupation and inefficient use of land. (2) Transfer system: Governments exert control, prohibiting free transfer and operating within urban-rural dual structure [13,31]. At present, rural residential land cannot be directly assigned, transferred, or leased, and rural land must be expropriated and converted into State land before it can be supplied to the market. This strengthens the welfare characteristics of rural land while weakening its asset characteristics and hinders rural migrants from realizing land added-value benefits. (3) Withdrawal system: Withdrawal means and compensation standards are not explicitly stipulated. According to the current law, rural collectives have the right to withdraw the farmers' residential base for public purposes but only have to pay the "reasonable" compensation.

### 2.3. Research Gap

Despite the extensive literature on China's land system and human–land relationship, there is relatively limited literature focusing on the linkage between the two, and those studies generally regard rural land system barriers as the key to the imbalance of human–land relations. There is a lack of in-depth studies on the detailed mechanisms of rural homestead system arrangements regarding this problem, especially from the perspective of multi-agent decision behavior, as well as a shortage of considerations on the bonds between the rural land system, the household registration system, and the social security system. Moreover, in terms of the study methods, most research involved the theoretical analysis of system history, features, and flaws. A few quantitative studies were limited to willingness surveys, lacking macro-level statistical analyses. Moreover, most of them did not identify the mechanisms of land systems on different agents, such as farmers and local

governments, and their interactions. There is still research space for in-depth analysis of the complex effects and the detailed mechanisms of the rural homestead system on rural human–land relationships.

To fill the gap, this paper integrates the rural land system arrangements, the micro-agent decision behavior, and the macro-evolution of human–land relationships into a unified framework. In this framework, we propose two main theoretical mechanisms of the current rural land system on human–land relationships: one is the land finance preference mechanism of governments, and the other is the land attachment mechanism of farmers' rights and interests. Then, we construct the baseline model and the spatial panel Durbin model for empirical study, using the panel data of the Yangtze River Delta (YRD) urban agglomeration during the period of 2010–2021. The findings may contribute to a better understanding of the role of rural homestead system reform in improving land use efficiency and promoting high-quality urbanization.

### 3. Conceptual Framework and Hypotheses

*3.1. Conceptual Framework*

To expose the mechanisms of the rural land system on the mismatch between rural people and rural land in detail, we need to construct a unified conceptual framework to incorporate the crucial factors into the human–land system. Rural human–land relationships are influenced by rural residents' action selection, as well as by local governments' policy choices under the land system arrangements. Based on the land function theory and public domain of property rights theory, this paper establishes the framework to demonstrate how the rural land system impact current uncoordinated human–land relations, as is shown in Figure 1. In this framework, we put forward two main mechanisms of the land system: (1) LFPM: In order to increase the financial income, local governments show a strong preference for not only pursuing the fiscal income of the rural land transfer revenue but also avoiding the fiscal cost for taking measures to help rural migrants transform into full citizens (indicated by the dashed line in Figure 1). (2) LAM: Owing to the insufficiency of rural social security, as well as the expectant land value-added interests, farmers form a strong attachment to their rural homesteads (indicated by the solid line in Figure 1). The two mechanisms lead to the imbalanced man–land relationship due to affecting the uncoordinated process of population urbanization and land urbanization. We discuss them in more detail in the next subsection on hypotheses.

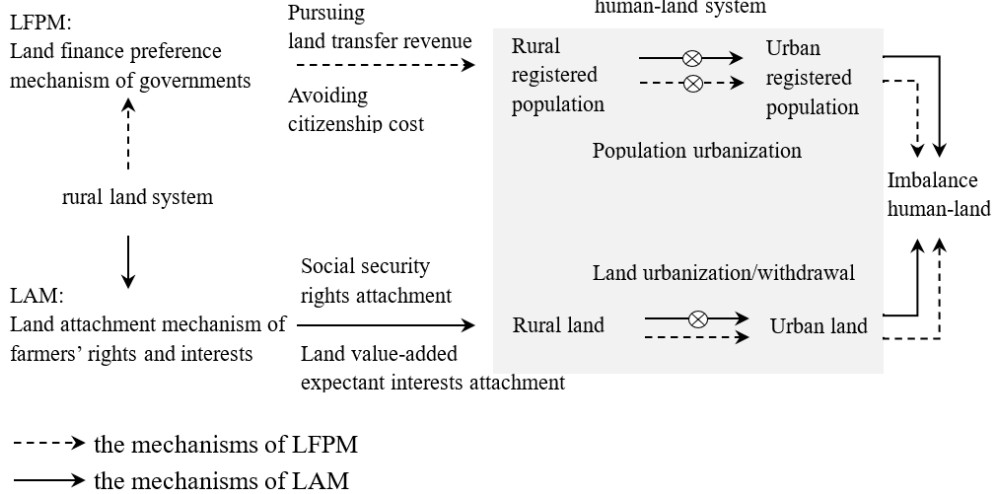

**Figure 1.** Conceptual framework of the mechanisms of the rural land system on the imbalance human–land relationship.

### 3.2. Hypotheses

Based on the land function theory and the public domain of property rights, this paper proposes two mechanism hypotheses of the effect of the rural land system on the rural imbalance of human–land relationships.

On the one hand, according to Barzel's theory of the public domain of property rights [32], the property rights of resources that are not defined clearly and completely form the "public domain" of property rights [33], which causes a benefit game and conflicts among various actors. This provides a good theoretical horizon for understanding the mechanisms of farmers' and local governments' decision-making behavior. China's rural land system arrangements are based on land administrative law, which is the basic law of land management in China, stipulating the basic system of rural land use, transfer, expropriation, compensation, and other aspects. China's current rural land system leads to a large public domain of rural land property rights:

(1) Property rights discrimination. Land administrative law clearly specifies that rural land must be converted into state-owned land through land expropriation by governments before it can legally enter the land market. This provision took effect on a national scale in 1999 and is still valid today. In practice, this has left the huge rural land value-added benefits in the "public domain", which have been seized by local governments with the dominant power [34]. Land finance is regarded as a notable fiscal phenomenon whereby local governments' fiscal revenue heavily depends upon land conveyance fees and land-related taxes. Due to land finance preferences, local governments tend to push rural land urbanization [35] while lacking the incentive to promote citizenship for rural migrants, which would incur a high fiscal cost. This leads population urbanization to drop far behind land urbanization and causes the imbalance of rural human–land relationships, giving birth to LFPM of local governments.

(2) Property rights restrictions. The land administrative law of China stipulates that rural land ownership belongs to rural collective organizations, and farmers only own the land-use rights, which are obtained with no compensation by their identity of rural collective membership [36]. The law also strictly restricts the land-use right, which can only be transferred within the collective members, and prohibits urban citizens and enterprises involved in the rural land-use right transaction. The above rules have been valid across the whole country since their birth until now. By restricting farmers' right to dispose of and benefit from land property rights, their land rights and interests remain in the "public domain of property rights" and lead to conflicts of rural land value-added benefit distribution among local government, rural collective organizations, and farmers [37]. The institutional rent-seeking space of the rural land system strengthened local governments' land finance preference mechanism. Moreover, this weakened farmers' (especially those who were rural migrants) exit willingness to idle rural residential land and strengthened the attachment of land value-added expectant benefits on their land-use right [38], which bring about the birth of LAM of farmers' value-added expectant interests.

On the other hand, according to the land function theory, the rural homestead has not only the basic functions of the land itself, such as the functions of living space and production space for human activities, but also its own unique functions, such as the functions of social security, political stability, and assets appreciation [39]. At different stages of urban–rural development, the functions of rural homesteads are constantly changing. Since China implemented the strict urban–rural dual household registration system over a long time, and owing to the inadequate rural social security system [40], rural homesteads play a critical role in keeping social security and political stability. As Chinese farmers' land-use rights are obtained with no compensation and for an undefined and long period of time, numerous farmers are strongly attached to their rights of social security and social welfare on their homesteads [41]. Although leaving the rural land, rural migrants do not give up their land-use rights easily when they face unemployment risks,

high living costs, unequal social welfare, and public services in the employment cities [42]. This leads to the increasingly serious phenomena of "fewer rural permanent residents, more idle residential land" and "hollow village", accelerating uncoordinated rural–human land relationships, which form the LAM of farmers' social security rights.

In summary, we put forward the following hypotheses:

**Hypothesis 1:** *the land attachment mechanism of farmers' rights and interests promotes the imbalance of rural human–land relations.*

**Hypothesis 2:** *the land finance preference mechanism of governments promotes the imbalance of rural human–land relations.*

The following text will further validate the hypotheses through econometric models.

## 4. Materials and Methods

### 4.1. Study Area

#### 4.1.1. Yangtze River Delta Urban Agglomeration in China

The YRD urban agglomeration covers the provinces and municipalities of Shanghai, Jiangsu, Zhejiang, and Anhui, with a total of 41 cities, an area of more than 350,000 square kilometers and a population of more than 230 million, making it the largest urban agglomeration in China, with the highest level of economic development. The cities account for about 3.7% of the country's land area, 16.7% of the country's total population, and 24.1% of the country's total economy.

In 2021, the GDP of 41 cities in YRD was CNY 27.7 trillion, accounting for 24.2% of the national GDP. Twenty-four cities in the whole country have a GDP exceeding CNY 1 trillion, and there are eight cities in YRD (Shanghai, Suzhou, Hangzhou, Nanjing, Ningbo, Wuxi, Hefei, and Nantong). The average value of the GDP of the 41 cities is CNY 675.1 billion, of which Shanghai's GDP (CNY 4321.5 billion) is much higher than the others; there are six cities in Jiangsu with a value higher than the average value (Suzhou, Nanjing, Wuxi, Nantong, Changzhou, and Xuzhou); there are four cities in Zhejiang higher than the average value (Hangzhou, Ningbo, Wenzhou, and Shaoxing); in Anhui, only Hefei has a higher-than-average value.

In the YRD region, urban development, in general, shows the characteristics of high in the east and low in the west, with Shanghai as the center and Hangzhou, Suzhou, Wuxi and Changzhou, Nanjing, Ningbo, and Hefei as sub-centers of the layout. The development is not balanced between cities. The city scale level covers mega, large, medium and small-sized cities, and the level of economic development covers the more developed and less developed cities. Taking the 41 cities in YRD as the study area can get a representative research sample set. The diagram of the YRD urban agglomeration is shown in Figure 2.

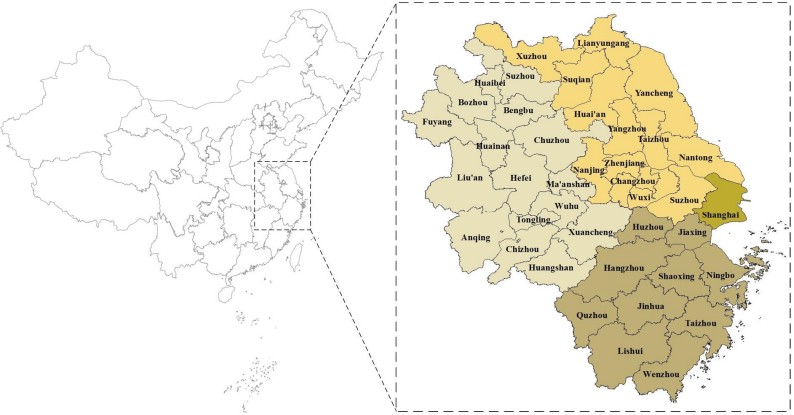

**Figure 2.** Diagram of Yangtze River Delta (YRD) urban agglomeration.

### 4.1.2. Characteristics of Rural Human–Land Relationships in the YRD Region

Theoretically, in the condition of no system barriers, urban–rural and human–land relationships should be coordinated with the mobility of population and land factors between rural and urban areas. However, in most of China's rural areas, the per capita residential land area has shown an increasing trend year by year. Based on officially published urban statistical yearbooks and national land use survey data, we measured the state of rural human–land relationships according to the following indicators: (1) the change of per capita residential land area and (2) the elasticity of land expansion to population growth by the Elasticity Value of Tapio Model, which is measured by Equation (1) [43]:

$$E\,land - pop = \frac{\Delta land}{\Delta pop} = \frac{(land_t - land_0)/land_0}{(pop_t - pop_0)/pop_0}, \tag{1}$$

where $E$ is the elasticity value; $pop_t$ and $pop_0$ are the size of the population in the final and initial years, respectively; and $land_t$ and $land_0$ are the volume of residential land in the final and initial years, respectively.

The results are as follows. From 2010 to 2021, the residential population in the rural areas of the YRD decreased by 22.2%, while the residential land area increased by 9.3%, and the residential land area per capita went up from 307.3 m$^2$ to 431.9 m$^2$, an increase of 40.5%. Figure 3 shows the elasticity of land expansion to population growth. More than 3/4 of cities in YRD are distributed in the second quadrant and in its vicinity, where the population is declining while the residential land area is increasing instead. Figure 4 shows the change rate of the resident population and the residential land area of 41 cities, which intuitively reflects the current serious problem of the imbalance of rural human–land relationship, that is, with rural residents migrating to cities for work, rural residential land did not cease to expand. This presents an important agenda for the sustainable development of rural areas in China. Consequently, the mechanisms of the rural land system on this problem are worth exploring further.

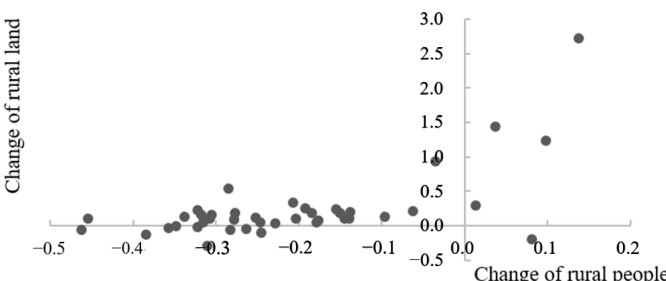

**Figure 3.** The change of rural human and rural residential land of 41 cities in 2021 vs. 2010.

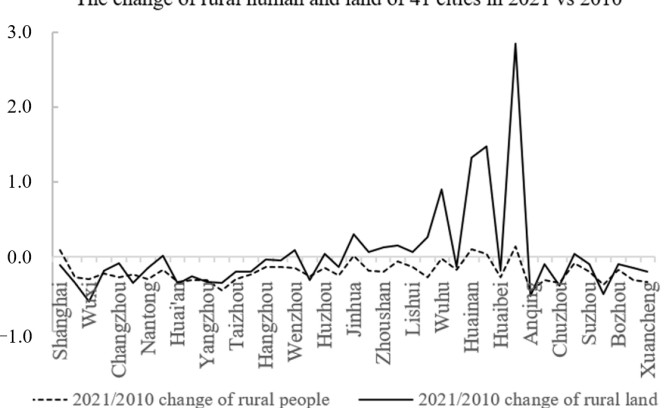

**Figure 4.** The change of rural human and rural residential land of 41 cities in 2021 vs. 2010.

### 4.2. Data Sources

This paper generated a panel data set of 41 cities in the YRD urban agglomeration from 2010 to 2021. The data for rural homestead areas were obtained from the National Land Use Survey Database of the Ministry of Natural Resources of China. The data for permanent population size, registered residence population size, per capita urban–rural disposable income, per capita urban–rural consumtion expense, net inflow rate of the urban population, and the proportion of secondary industry GDP were obtained from the 2011–2022 City Statistical Yearbook of each city. The data for land transfer income were collected from the municipal finance bureau and auditing bureau of each city. All data are publicly available.

### 4.3. Regression Models

#### 4.3.1. Variables Selection

Explained variable ($Y_{it}$): $Land_{it}$ is the amount of rural homestead area per capita, which is selected as a proxy variable for the state of rural human–land coordination.

Explanatory variables ($X_{it}$): $permanent\ urbaniz_{it}$ is the urbanization rate of permanent population, namely, the proportion of people who have been living in a specific city's urban area for more than 6 months. $registered\ urbaniz_{it}$ is the urbanization rate of the registered residence population, namely, the proportion of people who own urban household registration. These two indicators were chosen as the proxy variables for system barriers, which refer to the system obstacles of urban–rural land and population factors' mobility. $land\ finance_{it}$ is the annual land transfer income of the local government.

Control variables ($Z_{it}$): $income_{it}$ and $expense_{it}$ are the proxy variables for urban–rural disparity. $pop\ inflow_{it}$ is the proxy variable for the economic prospects of the city. $GDP_{it}$ is the proportion of secondary industry GDP in the whole GDP, chosen as the proxy variables for economic structure. Table 1 shows the list of variables and proxy indicators.

**Table 1.** Variables and the proxy indicators.

| Variables | | The Proxy Indicators |
|---|---|---|
| Rural man–land relationship | $land_{it}$ | Amount of per capita rural homestead area |
| System barrier | $permanent\ urbaniz_{it}$ | Urbanization rate of permanent population |
| | $registered\ urbaniz_{it}$ | Urbanization rate of registered residence population |
| Land finance | $land\ finance_{it}$ | Annual land transfer income of the local government |
| Urban–rural disparity | $income_{it}$ | Ratio of per capita urban–rural disposable income |
| | $expense_{it}$ | Ratio of per capita urban—rural consumption expense |
| Economic prospect | $pop\ inflow_{it}$ | Net inflow rate of urban population |
| Economic structure | $GDP_{it}$ | Proportion of secondary industry GDP |

#### 4.3.2. Descriptive Statistics

This study built a dataset of the related variables of 41 cities in the YRD urban agglomeration from 2010 to 2021. Table 2 presents the descriptive statistics of variables.

**Table 2.** Descriptive statistics of variables.

| Variables | N | Mean | Sd | Min | Max |
|---|---|---|---|---|---|
| $land_{it}$ | 492 | 202.77 | 62.32 | 57.54 | 401.26 |
| $permanent\ urbaniz_{it}$ | 492 | 61.03 | 12.25 | 29.10 | 89.60 |
| $registered\ urbaniz_{it}$ | 492 | 49.26 | 26.00 | 9.91 | 89.60 |
| $land\ finance_{it}$ | 492 | 389.09 | 632.19 | 5.49 | 7276.93 |
| $income_{it}$ | 492 | 2.20 | 0.45 | 1.60 | 5.27 |
| $expense_{it}$ | 492 | 2.09 | 1.13 | 0.40 | 15.43 |
| $pop\ inflow_{it}$ | 492 | −0.01 | 0.22 | −1.36 | 0.64 |
| $GDP_{it}$ | 492 | 47.23 | 7.56 | 26.49 | 74.73 |

4.3.3. Model Specification

Equations (2) and (3) represent the panel model and the spatial Durbin model, respectively.

$$Y_{it} = a + \beta X_{it} + \mu_i + \varepsilon_{it} \tag{2}$$

$$Y = \rho WY + \beta X + \theta WX + \varepsilon \quad (\varepsilon = \lambda W\varepsilon + \mu) \tag{3}$$

In Equations (2) and (3), $i$ denotes the sample city, $t$ denotes the year, $Y$ and $X$ are the explained and explanatory variables, respectively. $\beta$ and $\theta$ denote the coefficient to be estimate; $W$ denotes the spatial weight matrix; $\rho$ denotes the spatial lag coefficients; and $\varepsilon$ denotes the random error term, $\lambda$ is the coefficient of the spatial residual term, and $\mu$ is the random disturbance term.

Considering the sample data characteristics, this paper sets the benchmark model as a dynamic panel model $M1$, adding the first-order lag term of the explained variables in the control variables to improve the goodness of fit of the model. $M2$, $M3$, $M4$, and $M5$ are robustness test models for the baseline model $M1$. $M2$ added the control variable $Public\ services_{it}$ to $M1$. $M3$, $M4$, and $M5$ use $land\ price_{it}$, $house\ sales\ income_{it}$ and $house\ price_{it}$, respectively, to replace the explanatory variable $land\ finance_{it}$. $M6$ added the cross-multiplier term of the explanatory variables $land\ finance_{it}$, $permanent\ urbaniz_{it}$, and $registered\ urbaniz_{it}$ to the baseline model $M1$ to test the interaction effects of LFPM and LAM. In addition, this paper established a spatial Durbin model $M7$ to study the spatial spillover effects of the explanatory variables. Formulas (4) to (10) show the model specifications.

$$M1: Land_{it} = a + \beta_1 land_{it}(-1) + \beta_2 permanent\ urbaniz_{it} + \beta_3 registered\ urbaniz_{it} + \beta_4 land\ finance_{it} \\ + \beta_5 income_{it} + \beta_6 expense_{it} + \beta_7 pop\ inflow_{it} + \beta_8 GDP_{it} \tag{4}$$

$$M2: Land_{it} = a + \beta_1 land_{it}(-1) + \beta_2 permanent\ urbaniz_{it} + \beta_3 registered\ urbaniz_{it} + \beta_4 land\ finance_{it} \\ + \beta_5 income_{it} + \beta_6 expense_{it} + \beta_7 pop\ inflow_{it} + \beta_8 GDP_{it} + \beta_9 Public\ services_{it} \tag{5}$$

$$M3: Land_{it} = a + \beta_1 land_{it}(-1) + \beta_2 permanent\ urbaniz_{it} + \beta_3 registered\ urbaniz_{it} + \beta_5 income_{it} \\ + \beta_6 expense_{it} + \beta_7 pop\ inflow_{it} + \beta_8 GDP_{it} + +\beta_{10} land\ price_{it} \tag{6}$$

$$M4: Land_{it} = a + \beta_1 land_{it}(-1) + \beta_2 permanent\ urbaniz_{it} + \beta_3 registered\ urbaniz_{it} + \beta_5 income_{it} \\ + \beta_6 expense_{it} + \beta_7 pop\ inflow_{it} + \beta_8 GDP_{it} + \beta_{11} house\ sales\ income_{it} \tag{7}$$

$$M5: Land_{it} = a + \beta_1 land_{it}(-1) + \beta_2 permanent\ urbaniz_{it} + \beta_3 registered\ urbaniz_{it} + \beta_5 income_{it} \\ + \beta_6 expense_{it} + \beta_7 pop\ inflow_{it} + \beta_8 GDP_{it} + \beta_{12} house\ price_{it} Y_{it} = a + \beta X_{it} + \mu_i + \varepsilon_{it} \tag{8}$$

$$M6: Land_{it} = a + \beta_1 land_{it}(-1) + \beta_2 permanent\ urbaniz_{it} + \beta_3 registered\ urbaniz_{it} + \beta_4 land\ finance_{it} \\ + \beta_5 income_{it} + \beta_6 expense_{it} + \beta_7 pop\ inflow_{it} + \beta_8 GDP_{it} + \beta_{13} land\ finance_{it} \\ * permanent\ urbaniz_{it} + \beta_{14} land\ finance_{it} * registered\ urbaniz_{it} \tag{9}$$

$$M7: Land_{it} = \rho WLand_{it} + \beta_1 land_{it}(-1) + \beta_2 permanent\ urbaniz_{it} + \beta_3 registered\ urbaniz_{it} \\ + \beta_4 land\ finance_{it} + \beta_5 income_{it} + \beta_6 expense_{it} + \beta_7 pop\ inflow_{it} + \beta_8 GDP_{it} \\ + \theta_1 Wland_{it}(-1) + \theta_2 Wpermanent\ urbaniz_{it} + \theta_3 Wland\ finance_{it} + \varepsilon_{it} \tag{10}$$

## 5. Results

*5.1. Baseline Regression Results*

All of the data were standardized before conducting the regression analysis. Table 3 shows the basic regression results and robustness checks of Model 1.

**Table 3.** Results of the baseline model and robustness checks.

| Variables | Baseline Model | | Robustness Checks | | |
|---|---|---|---|---|---|
| | M1 | M2 | M3 | M4 | M5 |
| $land_{it}\,(-1)$ | 0.704 *** | 0.698 *** | 0.716 *** | 0.681 *** | 0.687 *** |
| | (0.035) | (0.035) | (0.035) | (0.035) | (0.036) |
| permanent urbaniz$_{it}$ | 0.407 *** | 0.411 *** | 0.394 *** | 0.433 *** | 0.438 *** |
| | (0.054) | (0.053) | (0.053) | (0.054) | (0.055) |
| registered urbaniz$_{it}$ | −0.058 *** | −0.062 *** | −0.062 *** | −0.05 *** | −0.048 *** |
| | (0.014) | (0.014) | (0.014) | (0.014) | (0.014) |
| land finance$_{it}$ | −0.101 ** | −0.095 ** | | | |
| | (0.04) | (0.04) | | | |
| income$_{it}$ | −0.164 *** | −0.172 *** | −0.167 *** | −0.166 *** | −0.164 *** |
| | (0.032) | (0.032) | (0.032) | (0.032) | (0.032) |
| expense$_{it}$ | 0.117 *** | 0.114 *** | 0.119 *** | 0.13 *** | 0.122 *** |
| | (0.038) | (0.038) | (0.038) | (0.038) | (0.038) |
| pop inflow$_{it}$ | −0.248 *** | −0.264 *** | −0.235 *** | −0.253 *** | −0.252 *** |
| | (0.049) | (0.049) | (0.049) | (0.048) | (0.048) |
| GDP$_{it}$ | −0.191 *** | −0.206 *** | −0.19 *** | −0.226 *** | −0.221 *** |
| | (0.03) | (0.031) | (0.03) | (0.032) | (0.032) |
| Public services$_{it}$ | | −0.042 ** | | | |
| | | (0.017) | | | |
| land price$_{it}$ | | | −0.103 ** | | |
| | | | (0.04) | | |
| house sales income$_{it}$ | | | | −0.14 *** | |
| | | | | (0.033) | |
| house price$_{it}$ | | | | | −0.128 *** |
| | | | | | (0.037) |
| constant | 0.226 *** | 0.263 *** | 0.273 *** | 0.238 *** | 0.237 *** |
| | (0.038) | (0.041) | (0.043) | (0.038) | (0.038) |
| FE | Yes | Yes | Yes | Yes | Yes |
| Observations | 492 | 492 | 492 | 492 | 492 |
| R-squared | 0.862 | 0.863 | 0.862 | 0.865 | 0.863 |

Note: Standard errors are in parentheses. ** and *** denote the significance levels of 0.05 and 0.01, respectively. FE denotes fixed-effects.

### 5.1.1. Significant Test

The estimation results of Model 1 in Table 3, column M1, show that all variables are statistically significant and reasonable.

First, we look at the two indicators of the system barrier variable, *permanent urbaniz$_{it}$* and *registered urbaniz$_{it}$*, the estimation results of which are consistent with the expected signs assumed by Hypothesis 1. (1) The coefficient of *permanent urbaniz$_{it}$* is 0.41 and is significant at the 1% level. The positive coefficient indicates that an increase in the urbanization rate of permanent residents leads to an increase in the per capita area of rural homesteads. Accordingly, although the number of permanent residents in rural areas decreases when a large number of rural migrant workers move to the city, the per capita rural homestead area increases rather than decreases. Many rural migrants have left the countryside but still keep the rural homesteads of their hometowns [44], leaving the land unused for most of the year. This has a positive effect on the growth of the rural homestead area per capita. (2) The coefficient of *registered urbaniz$_{it}$* is −0.06, with a significance at the 1% level. This means that for every 0.06 standardized unit increase in the urbanization rate of the registered population, the rural homestead area per capita decreases by 1 unit. This suggests that if measures are taken to reduce barriers in the household registration system and promote the citizenship of rural migrant workers [45], this will accelerate their decision to leave rural homesteads.

Next, we look at the indicators of the land finance variable. The coefficient of *land finance$_{it}$* is −0.1, with a significance at the 5% level, of which the estimation results are consistent with the expected signs that Hypothesis 2 assumes. This means that

for every 0.1 standardized unit increase in local government land transfer revenue, the per capita rural homestead area decreases by 1 unit. This provides evidence that the process of population urbanization has lagged far behind land urbanization due to local governments' preference for land finance.

Finally, we look at the control variables: (1) the coefficients of $income_{it}$ (per capita urban–rural disposable income gap), $pop\,inflow_{it}$ (net urban population inflow rate), $GDP_{it}$ (economic structure) are $-0.16$, $-0.25$, and $-0.19$, respectively, and are significant at the 1% level. The results indicate that higher labor income, greater population attractiveness, better economic prospects, and the structure of the city have positive effects on the decision of rural migrant workers to leave rural homesteads. (2) The coefficients of $expense_{it}$ (per capita urban–rural consumption expenditure gap) are 0.12 and are significant at the 1% level, indicating that higher urban living cost hinders rural homesteads exit.

### 5.1.2. Robustness Checks

Model $M2$ added the control variable $Public\,services_{it}$, which is the number of doctors per 10,000 inhabitants. Models $M3$, $M4$, and $M5$ replaced the explanatory variable $land\,finance_{it}$ with $land\,price_{it}$, $house\,sales\,income_{it}$, and $house\,price_{it}$, respectively. The robustness test results of Model 1 in Table 3, columns M2–M5, show that the baseline model estimates are robust.

### *5.2. Mechanism Analysis*
### 5.2.1. LAM of Farmers Promotes the Imbalance of Rural Human–Land Relationships

The regression results of the variables $permanent\,urbaniz_{it}$ (0.41) and $registered\,urbaniz_{it}$ ($-0.06$) in Model 1 show the effect of LAM on per capita rural homestead areas. This confirms Hypothesis 1: LAM promotes the imbalance of rural human–land relationships.

Specifically, the logic of LAM is as follows: LAM worked in the context of the current rural homestead system arrangements, which provide the system foundation for the attachment of the farmer's social security rights and asset appreciation expectant interests with their land use right. With the continuous reform of the dual urban-rural household registration system, the current rural land system has become the most critical system barrier to the coordination of rural people-land relations. On 30 July 2014, the opinions of the state council on further promoting the reform of the household registration system proposed to unify the urban-rural household registration system nationwide, abolish the distinction between agricultural and non-agricultural household registration, and fully implement the residence permit system. Since China's founding, the strict barriers of the dual urban-rural household registration system have begun to loosen, marking the first step towards equal social security rights for rural migrants.

However, the strong attachment of migrant farmers' rights and interests to rural land will not change much as long as the current rural homestead system remains in place [46]. This is because, in addition to the attachment of farmers' social security rights to land, farmers' land value-added interests are strongly attached to their rural homestead land use right [41]. With the improvement of farmers' property rights awareness, they would not withdraw from rural land without a satisfactory price or reasonable compensation [47]. In contrast, many rural migrant workers choose to transfer the wealth they have accumulated in the cities back to the countryside to expand their rural homesteads and rebuild houses [48]. This pattern of "spatial separation of man and land" is clearly the combined effect of the land attachment of farmers' social security rights and the expectation interests of assets appreciation [49], which promotes the unbalanced relationship between rural people and land.

In addition, the growing market demand for rural homesteads and houses strengthens the land attachment of farmers' property interests. In regions with an external population influx, a large number of external floating workers have gathered in urban villages and peri-urban villages. Moreover, some urban residents tend to live in rural areas for leisure and a better ecological environment. They have demands to buy or rent rural residential

land and houses. Although the current rural land system strictly restricts the transfer of land outside the members of rural collectives, some rural collectives and farmers, while motivated by huge interests in land appreciation, illegally expand rural construction land and develop commercial housing for rent or sale, giving rise to the huge "grey market" in rural land and house transactions [50]. Therefore, it is not surprising that the number of rural homesteads continues to increase as the urbanization rate of permanent urban residents increases. It is obvious that the land attachment of farmers' property interests leads to uncoordinated rural human–land relationships.

### 5.2.2. LFPM of Local Governments Promotes the Imbalance of Rural Human–Land Relationships

The regression results of the variables $land\ finance_{it}$ ($-0.1$) and $permanent\ urbaniz_{it}$ (0.41) in Model 1 show the effect of LFPM on the per capita rural homestead area. This verifies Hypothesis 2: LFPM promotes the imbalance between rural people and land.

In detail, the logic of LFPM is as follows: One of the most prominent features of China's current land system is the urban–rural dual structure. Farmers' land use rights face systemic discrimination due to the current rural land system. State-owned construction land use rights can be disposed of through sale, transfer or mortgage, while rural collective construction land use rights are subject to strict restrictions and require expropriation procedures before they can be converted into State-owned construction lands and put on the market. This creates a monopoly of local governments over the supply of land to the market. Land finance, i.e., transfer revenue and tax from land transactions, is deeply rooted in the monopoly status of governments. With rapid urbanization and the flourishing development of the real estate industry, land finance has become one of the most important sources of local fiscal revenue since the 2000s. From 2011 to 2021, national land transfer revenue increased from CNY3.3 trillion to CNY8.7 trillion, of which the proportion of total local fiscal revenue increased from 63% to 78% [51]. It is clear that the dual urban–rural land system arrangements form the institutional basis for FP Mand provide rent-seeking opportunities for the local government interest group.

On the one hand, local governments have a strong preference for increasing their land-transfer revenue [52]. Since the reform of the tax-sharing system and decentralization in 1994, the revenue capacity of local governments has decreased, and their expenditure responsibilities have increased, leading to the challenge of local budget deficits. As land is the most valuable asset controlled by local governments, they take land finance as the critical fiscal revenue source. In order to obtain fiscal capital to accelerate economic growth and provide public services, local governments rely on land development to generate land finance revenue [53], thereby accelerating excessive rural–urban land conversion. Due to political performance and promotion incentives, local governments have a strong willingness to obtain construction land quota and increase land transfer revenue through various means [54], such as rural land expropriation [55], village consolidation [56], rural land preparation and centralization [57], and the policy of "increasing and decreasing balance" between urban and rural construction land, that is, balancing the increase in urban construction land with the parallel increase in rural farmland, thereby reducing rural construction land [58]. As per the regression results of Model 1 shown in Table 3, the effect of $land\ finance_{it}$ on per capita rural homesteads is $-0.1$. In this process, it promotes rural land urbanization and the intensive use of rural residential land to some extent.

However, most of the added-value benefits of land urbanization are taken away by local governments [59]. Research shows that in the distribution of land revenue, farmers get only 5–10%, village collectives get 25–35%, and local governments get 60–70% [60]. Moreover, it should be noted that this local government-led mode of land-centered urbanization and rural homestead withdrawal generally occurs in a small part of suburban rural areas or urban villages with locational advantage. The farmers who become urban citizens through this method are also limited to a very small part of the whole farmers' group. The rural homesteads, in the vast majority of non-suburban rural areas, still remain

"dormant assets" that are not allowed to enter the land market [58], fueling the problem of the inefficient use of rural land in the wider countryside.

On the other hand, local governments have the motivation to lighten their financial burden. They would like to avoid the fiscal cost by taking few actions to help rural migrants transform into full citizens. Local governments have been faced with large fiscal gaps since the 1994 tax-sharing system reform and decentralization process. As the government is the dominant provider of social security and public services, the fiscal pressure of governments responsible for providing equal security and services to rural migrants is a crucial factor limiting citizenship development. In the past years of industrialization and rapid urbanization, because of the citizenship cost exclusion, local governments' actions to promote rural migrants turning to new urban citizens are far behind in their efforts to promote rural land urbanization. It is in accordance with the regression result of Model 1 shown in Table 3: The effect of *land finance*$_{it}$ on per capita rural homesteads is −0.1, which is much smaller than that of *permanent urbaniz*$_{it}$ (0.4). Local governments lack the willingness to bear the fiscal costs of providing equal social security and public services to rural migrants, which need to increase investment on education, healthcare, housing, job training, transportation, and so on. Moreover, the fiscal expenditure arrangements of local governments are tilted toward cities in the long run, neglecting the need for rural development. These lead to the current unbalanced and unsustainable rural human–land relationship.

### 5.3. Moderating Effect

To further clarify the relationships of the two mechanisms above, LAM and LFPM, this paper conducted a moderating effect test by adding *land finance*$_{it}$ ∗ *permanent urbaniz*$_{it}$ and *land finance*$_{it}$ ∗ *registered urbaniz*$_{it}$—the interaction term of the explanatory variables—into the baseline model M1. Table 4 and Figures 5 and 6 show the results of the moderating effect. In Figure 5, it can be seen that *land finance*$_{it}$ has an enhancement effect on *permanent urbaniz*$_{it}$'s positive influence of the per capita rural homestead area. In addition, in Figure 6, it shows that *land finance*$_{it}$ has an enhancement effect on *registered urbaniz*$_{it}$'s negative impact of the per capita rural homestead area. The results of the moderating effect indicate that the two mechanisms, LAM and LFPM, can reinforce each other.

**Table 4.** Results of the moderating effect.

| Variables | M1: Baseline Model | | M6: Moderating Effect | |
|---|---|---|---|---|
| | Coefficient | Ro S.E. | Coefficient | Ro S.E. |
| *land*$_{it}$ (−1) | 0.704 *** | 0.035 | 0.698 *** | 0.035 |
| *permanent urbaniz*$_{it}$ | 0.407 *** | 0.054 | 0.299 *** | 0.069 |
| *registered urbaniz*$_{it}$ | −0.058 *** | 0.014 | −0.003 | 0.029 |
| *land finance*$_{it}$ | −0.101 ** | 0.04 | −0.616 * | 0.35 |
| *income*$_{it}$ | −0.164 *** | 0.032 | −0.171 *** | 0.032 |
| *expense*$_{it}$ | 0.117 *** | 0.038 | 0.108 *** | 0.039 |
| *pop inflow*$_{it}$ | −0.248 *** | 0.049 | −0.247 *** | 0.049 |
| *GDP*$_{it}$ | −0.191 *** | 0.03 | −0.237 *** | 0.044 |
| *land finance*$_{it}$ ∗ *permanent urbaniz*$_{it}$ | | | 0.987 ** | 0.43 |
| *land finance*$_{it}$ ∗ *registered urbaniz*$_{it}$ | | | −0.437 ** | 0.214 |
| *constant* | 0.226 *** | 0.038 | 0.279 *** | 0.053 |
| FE | Yes | | Yes | |
| Observations | 492 | | 492 | |
| R-squared | 0.862 | | 0.864 | |

Note: Standard errors are in parentheses. *, **, and *** denote the significance levels of 0.1, 0.05 and 0.01 respectively. FE denotes fixed-effects.

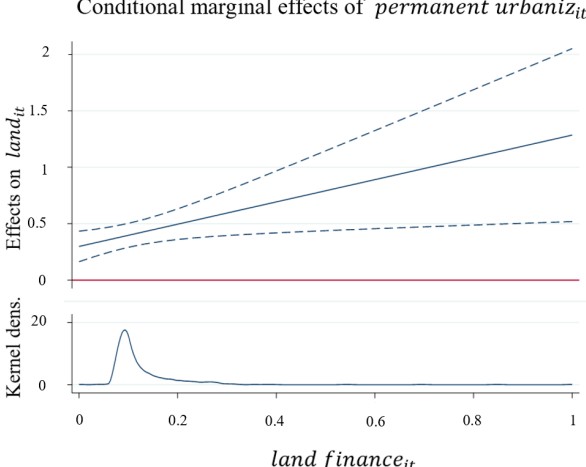

**Figure 5.** Diagram of the moderating effect.

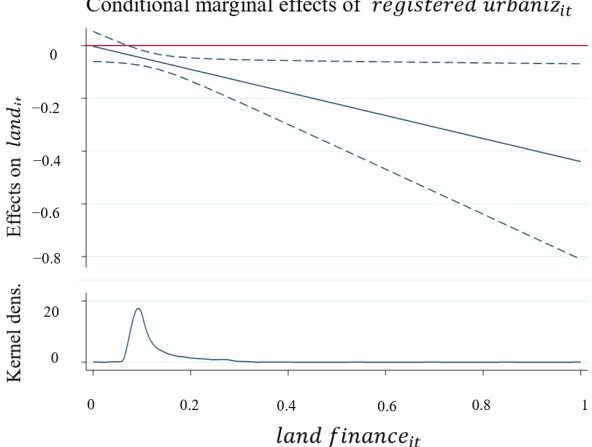

**Figure 6.** Diagram of the moderating effect.

Local governments, as the vested interests of the existing rural land system arrangement, have long depended on land finance [61]. Owing to land financial incentives, they not only lack the willingness to promote deep reform to break down the land system barriers at present and provide equal social security and welfare to rural migrants but also even intentionally set up rules to strengthen their own monopoly situation in the land transfer market and prohibit the huge scale of rural construction land directly entering the land market before it is transformed to State-owned land. Accordingly, without permanent residences in cities, when faced with the high living cost, unequal social services, and unemployment risk [62], rural migrants find it hard to settle in cities and hope that they can return to rural residences if they lose their jobs. Consequently, they have to attach their social security rights and property interests deeply to their rural land use rights, which are absolutely protected by the law. At the same time, to show off the decent life brought about by their urban experience, many rural migrants are willing to build larger houses in their hometowns once they earn money in the city. Therefore, we can see the increasingly serious phenomenon of "fewer rural residents, more rural residential land" under the mutual reinforcement of FP Mand LAM.

### 5.4. Spatial Spillover Effect

The previous literature suggests that there may be spatial correlations between some of the variables in the baseline Model 1 [63]. This paper constructed a dynamic spatial Durbin model M7, with the geographical neighborhood matrix as the spatial weight matrix, to further study the spatial spillover effect of related variables.

First, as the Moran's I index is 0.35 and its *p*-value is 0.00, it is suitable to apply the spatial Durbin model. Table 5 shows the results of the spatial spillover effect. We can see that the spatial lag coefficients $\rho$ is significant, indicating that there is a spatial spillover effect in Model 7. Further, the regression results of the spatial lag variables show that the two mechanisms LAM and LFPM have a spatial spillover effect. (1) *permanent urbaniz$_{it}$*: its total effect on the per capita rural homestead area is 0.77, of which the direct effect is 0.41 and the indirect effect is 0.36. It means that the spatial spillover effect from *permanent urbaniz$_{it}$* of "neighboring cities" on the observed city's per capita rural homestead area is 0.36. (2) *land finance$_{it}$*: its total effect on the per capita rural homestead area is −0.28, of which the direct effect is −0.1 and the indirect effect is −0.18. It shows that the spatial spillover effect from *land finance$_{it}$* of "neighboring cities" on the observed city's per capita rural homestead area is −0.18.

**Table 5.** Results of the spatial spillover effect.

| Variables | M1: Baseline Model | | M7: Spatial Spillover Effect | |
|---|---|---|---|---|
| | Coefficient | Ro S.E. | Coefficient | Ro S.E. |
| $land_{it}$ (−1) | 0.704 *** | 0.035 | 0.673 *** | 0.033 |
| *permanent urbaniz$_{it}$* | 0.407 *** | 0.054 | 0.410 *** | 0.062 |
| *registered urbaniz$_{it}$* | −0.058 *** | 0.014 | −0.036 * | 0.019 |
| *land finance$_{it}$* | −0.101 ** | 0.040 | −0.104 *** | 0.037 |
| *income$_{it}$* | −0.164 *** | 0.032 | −0.183 *** | 0.030 |
| *expense$_{it}$* | 0.117 *** | 0.038 | 0.084 ** | 0.035 |
| *pop inflow$_{it}$* | −0.248 *** | 0.049 | −0.251 *** | 0.047 |
| $GDP_{it}$ | −0.191 *** | 0.030 | −0.192 *** | 0.037 |
| *constant* | 0.226 *** | 0.038 | | |
| $W_X$ : $land_{it}$ lag(1) | | | −0.153 * | 0.084 |
| $W_X$ : *permanent urbaniz$_{it}$* | | | 0.363 *** | 0.104 |
| $W_X$ : *land finance$_{it}$* | | | −0.181 ** | 0.077 |
| *Spatial* : $\rho$ | | | 0.139 ** | 0.066 |
| FE | Yes | | Yes | |
| Observations | 492 | | 492 | |
| R-squared | 0.862 | | 0.864 | |

Note: Standard errors are in parentheses. *, **, and *** denote the significance levels of 0.1, 0.05 and 0.01 respectively. FE denotes fixed-effects.

On account of the spatial spillover effect of the two mechanisms LAM and LFPM on the imbalance of rural human–land relations, it is necessary to pay attention to the synergy effect of cities in urban agglomerations when deepening system reforms.

## 6. Discussions and Conclusions

### 6.1. Discussions

In this section, we make further discussions on the possible contributions and limitations of this paper and present the issues that still need to be further studied in the future.

First, this study contributes to filling in some of the gaps in the literature. The findings verified that the current rural land system arrangements have a negative effect on the coordination of rural human–land relationships, which is also supported by the previous literature [13,48,54,61]. However, this paper is distinguished from these studies by the further exploration of the detailed mechanisms and their complex effects. To be specific, this paper further verified that the rural land system affects rural human–land relationships through two main mechanisms: LAM and LFPM. In particular, owing to the rural land property rights discrimination and restrictions of the current rural land system, farmers attach deeply to rural homesteads (LAM), and local governments show strong land finance preference (LFPM), which both lead to the imbalanced mobility of population and land factors between rural areas and urban areas. Furthermore, it helped fill the gaps in the

insufficient mechanism analyses and quantitative studies. Moreover, this paper found that local governments' land finance preference intensifies the attachment of farmers' rights and interests to land, making up for the research gap in the mechanisms' interaction effects. In addition, this paper verified that LAM and LFPM are spatially correlated by conducting the dynamic spatial Durbin model regression, making up for the gap in the mechanisms' spatial spillover effects.

Second, there are also some limitations to this paper, as follows:

(1) The empirical analysis was conducted on the basis of macro-level data, which cannot accurately and fully reflect the agents' decision-making mechanisms of farmers and local governments in the context of the current rural land system.

(2) Although this paper considers the linkage of the rural homestead system and the household registered system and the impact of this linkage on human–land relationship in theory, there is a lack of econometric analysis to distinguish and compare the effect of the two critical system.

(3) At present, the reform pilot work on the current rural homestead system is in process in more than 100 counties in China. Under the premise of ensuring rural social stability, exploring how to promote the market allocation of rural land resources is one of the most important objectives of the pilot work. On this issue, the central government encourages the moderate deregulation of rural homestead withdrawal and transfer. The reform pilot areas provide materials and data to compare the allocation effect of land resources between the previous fair allocation method and the market allocation method. This may help to provide beneficial implications for optimizing land resource allocation.

(4) The mechanism analysis of this article is limited to the formalized behavioral mechanism analysis framework, which focuses on of how the private group and the public local administration take actions to maximize economic benefits. However, beyond the economic benefits, both the private agents and public agents also consider social benefits and environmental benefits when making behavioral choices. This means that there are not only the flow of land and population factors from rural areas to urban areas but also the flow of resources and factors from urban areas to rural areas that provide meaningful implications to reform the current urban-centered planning policies and the rural land system. This paper did not take social and environmental reasons into consideration.

We hope to explore these issues further in future studies.

*6.2. Conclusions*

At present, with rural residents continuing to decline, the rural homestead area increases rather than decreases, causing the serious problem of imbalanced rural human–land relationships in the vast Chinese rural areas. With the reform of the household registration system in 2014, rural land system arrangements in China are regarded as the critical system root of this problem. Rural land property rights restrictions and discriminations affect the coordination movement of population and land factors between rural areas and urban areas. This paper incorporated the rural land system, farmers' action selection, local governments' policy choice, and rural human–land relationships into a unified framework and proposed two mechanism hypotheses based on land function theory and public domain of property rights theory: one is LAM, the other is LFPM. Then, it further empirically discussed the mechanisms and effects using data from 41 cities in the Yangtze River Delta from the period of 2010–2021. The main conclusions are drawn as follows:

(1) LAM promotes the imbalance of rural human–land relationships. On the one hand, rural migrants who are in the state of semi-urbanization have to attach their social security rights and interests to the land use rights of hometown rural homesteads, which are obtained with no compensation and indefinite duration. On the other hand, owing to the rural land transfer restrictions and discrimination, rural migrants have

to attach their property expectant interests deeply to rural homesteads. These make rural migrants leave the countryside without withdrawing from the land.

(2) LFPM drives the imbalance of rural human–land relationships. The dual urban–rural land system provides local governments with the privilege of monopolizing the supply in the land transfer market. This way, they capture most of the value-added benefits from land urbanization. Due to the land finance incentive, local governments not only have the land transfer revenue preference but also have the cost exclusion for rural migrants' citizenship. These make the urbanization of the rural population lag far behind rural land urbanization and negatively affect the coordination of rural human–land relationships.

Moreover, the moderating effects suggest that LFPM can strengthen the effect of LAM, and the spatial Durbin model results show that both LAM and LFPM have spatial spillover effects. Accordingly, the synergy effect of cities in urban agglomerations needs to be considered when deepening land system reform.

It is hoped that these findings will provide policy guidance for solving the problem of unbalanced rural human–land relationships. The most critical path is to deepen the rural land system reform to break the system barrier of free movement of population, land, and capital factors between rural and urban areas. Specifically, (1) establish the compensation system for rural land use and land withdrawal and respect the willingness and subject status of farmers; (2) establish the unified urban and rural land market, improve the market allocation mechanism of rural land resources, allow for rural collective-owned land to be allocated, leased, or invested in shares, and ensure that rural land enters the market with the same rights and the same price as State-owned land; (3) improve the distribution mechanism of land value added-benefits, balance the interests of the government, rural collective organizations, and farmers, and give priority to guaranteeing farmers' rights; and (4) deepen the reform of the linkage of the dual urban–rural land system, the household registration system, and the social security system.

**Author Contributions:** Conceptualization, Y.Y. and F.H.; data curation, Y.S. and K.D.; formal analysis, F.H.; investigation, Y.Y. and K.D.; methodology, Y.Y. and Y.S.; software, Y.S.; validation, Y.Y.; visualization, Y.S.; supervision, F.H. and K.D.; writing—original draft preparation, Y.Y., F.H. and Y.S.; writing—review and editing, Y.Y. and K.D.; All authors have read and agreed to the published version of the manuscript.

**Funding:** National Social Science Foundation of China (23CJY017).

**Data Availability Statement:** The data presented in this study are available on request from the corresponding author. The data are not publicly available due to privacy restrictions.

**Conflicts of Interest:** The authors declare no conflicts of interest.

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
