# Peer review of "The Effects and Mechanisms of the Rural Homestead System on the Imbalance of Rural Human–Land Relationships: Evidence from the Yangtze River Delta Urban Agglomeration in China"

_land, doi:10.3390/land13020137_

Round 1

Reviewer 1 Report

Comments and Suggestions for Authors

The article provides relevant insights and sheds light on the biases of the current dual urban-rural land regime in China along with its distortive effects in term of effective land use and occupation.

The methodology adopteed to demonstrate te joint effects , under this regard, of the dual and complementary mechanism on behalf of rural resident and the Local administration to catch and protect the various benefits deriving from the land ownership  and tenure rights  it sound and fitting with the scope. 

The referee only suggests some minor integrations or additional short reflections about the following points: 

1.     For a reader not deep expert of land regime management and of the institutionally framed urban – rural relationship in China deciphering the various categories adopted by the article it is not easy. Maybe a short introductory section or some refinements about the various categories or concept could be helpful (e.g. citizenship tax exclusion mechanism); 

2.     The article adopts a strong formalized method drawing on the presume role of self interested and market driven behaviors on behalf either of private group and public local administration. Otherwise probably some other reasons that deserves at least mention affect the behaviors of both urban and rural resident that relates to the search for a healthier and safe and social living environment compared with the urbanized life condition in the metro and megalopolitan context. That could feed some integrative reflections not exclusively toward a market led land regime but also aimed to suggest some alternative planning measures and models aimed to reduce and, as much as possible, to revise current urbanization -and urban centered- policies by enhancing the inherited interesting polycentric rural homestead system;

3.     It is interesting how, in a so strictly ruled and managed land regime system, the article accounts for some form of “gray land market” on behalf of rural collectives and individual. Maybe also this element could be considered as an implicit reform demand but as well as an example of a possible outcome of a land regime reform totally left to the market that, also drawing on the experience of many western countries, doesn’t fit -and mainly fails- with the fair allocation of scarce resources and public goods as int the case of the land. 

Row 316.    Error indication

Row  391.  Bellow: to be verified

Comments on the Quality of English Language

 a revision on behalf of a mother tongue reader is suggested

Author Response

We greatly appreciate your professional suggestions on the manuscript. Please see the attachment.

Reviewer 2 Report

Comments and Suggestions for Authors

The content and sections of the article have been prepared in detail within the scope of the purpose of the study. Explanations can be simplified a little more to provide more understandable integrity, between methodology and result sections. The content and sections are organized in sufficient detail. The results have been discussed well.

Author Response

(The authors gave the same response as above.)

Reviewer 3 Report

Comments and Suggestions for Authors

The paper's topic is very important. However, some issues need to be addressed carefully. Please see my comments and suggestions below.

The article should include  research objective in papier and the abstract. Please complete.

Verse 315 has an error- please correct.

Table 5 please move behind the appendix.

,,Property rights discrimination and Property rights restrictions”  This is a very interesting topic.   Do these topics include the whole country, or only local governments can implement this policy. Please add 2-3 sentences on this topic, as the international reader has no knowledge on this subject.

Is this an authored research proposal? If yes, please add a sentence about it? What is innovative about the research?  What distinguishes the research conducted?

Please change the order of first the Discussion then the Conclusion.

*In the conclusion

*Please indicate what is the added value of the research conducted.

In conclusion the article is interesting. The reviewer has never been to China, but the article gave an understanding of The Effects and Mechanisms of the Rural Homestead System on the Imbalance of Rural Human-Land Relationship in the China .Thank you.

Author Response

(The authors gave the same response as above.)
